# Prognostic Value of Serum Transferrin Level before Radiotherapy on Radio-Sensitivity and Survival in Patients with Nasopharyngeal Carcinoma

**DOI:** 10.3390/jpm13030511

**Published:** 2023-03-13

**Authors:** Yuping Zhan, Li Su, Qiaojing Lin, Xiaoxian Pan, Xiaoxia Li, Weitong Zhou, Weijian Zhang, Jinsheng Hong

**Affiliations:** 1Department of Radiotherapy, First Affiliated Hospital of Fujian Medical University, Chazhong Road #20, Fuzhou 350005, Chinazwj9090@126.com (W.Z.); 2National Regional Medical Center, Binhai Campus of the First Affiliated Hospital, Fujian Medical University, Fuzhou 350212, China; 3Key Laboratory of Radiation Biology of Fujian Higher Education Institutions, The First Affiliated Hospital, Fujian Medical University, Fuzhou 350005, China

**Keywords:** nasopharyngeal carcinoma, transferrin, intensity-modulated radiotherapy, radio-sensitivity, survival

## Abstract

Purpose: To investigate the prognostic value of serum transferrin (TRF) level before intensity-modulated radiation therapy (IMRT) on radio-sensitivity and overall survival (OS) in patients with nasopharyngeal carcinoma (NPC). Methods: From October 2012 to October 2016, a total of 348 patients with NPC in the First Affiliated Hospital of Fujian Medical University were retrospectively analyzed in our study. The concentration of serum TRF was detected by the method of enzyme-linked immunosorbent assay (ELISA). In the whole group, 46 patients received IMRT, and 302 patients received IMRT plus chemotherapy. The radio-sensitive tumor was defined when the local tumor lesions disappeared completely in the nasopharyngeal MRI scan and no tumor residues were found under the electronic nasopharyngoscope one month after the end of radiotherapy. Results: The serum TRF level before IMRT was (1.34–3.89) g/L, with a median of 2.16 g/L and a mean of (2.20 ± 0.42) g/L. In the whole group, 242 cases (69.5%) were radiosensitive, and 106 cases (30.5%) were insensitive. The number of radiosensitive patients in the group of HTRF (transferrin > 2.16 g/L) and LTRF (transferrin ≤ 2.16 g/L) before radiotherapy was 129 (74.6%) and 113 (64.6%), respectively. The difference in radio-sensitivity between the two groups was statistically significant (*χ*^2^ = 4.103, *p* = 0.043). Logistic regression analysis showed that the level of TRF before radiotherapy (*OR* = 1.702; 95% *CI* 1.044~2.775; *p* = 0.033) was an independent factor for radio-sensitivity. The log-rank test showed that patients in the LTRF group achieved a significantly worse OS (*χ*^2^ = 5.388, *p* = 0.02) than those in the HTRF group. Cox regression analysis showed that baseline TRF level (*HR* = 1.706; 95% *CI* 1.065~2.731; *p* = 0.026) was an independent prognostic factor for overall survival. Conclusions: The low level of TRF before IMRT is a risk factor for radio-sensitivity and a prognostic factor for poor OS in NPC patients. It may be a promising marker to predict radio-sensitivity and OS in NPC patients who accept IMRT.

## 1. Introduction

The incidence of nasopharyngeal carcinoma (NPC) exhibits a remarkable regional disparity, with its high incidence in Asia, especially in the southern provinces of China, including Hong Kong, Guangdong, Guangxi, Fujian, and Jiangxi Province [1]. With the development of radiotherapy technology, intensity-modulated radiotherapy (IMRT) has become the primary method used to treat NPC patients, and patients in stage II–IV also received chemotherapy [2,3]. EBV virus infection and TNM stage are traditionally considered to be important factors affecting the prognosis of patients [4,5,6]. However, studies have shown that the clinical outcomes can vary widely among patients who have the same level of EBV DNA, TNM stage, pathological characteristics, and are treated with similar therapies. The difference in tissue radio-sensitivity is one of the main reasons [7]. For NPC patients, those who are resistant to radiotherapy are prone to local residual, which increases the probability of tumor recurrence and metastasis and consequently affects the quality of life and survival of patients. Therefore, studies focusing on the search for predictors of tissue or regression-based radio-sensitivity are becoming a focus in clinical and basic studies for NPC.

The tissue radio-sensitivity of NPC is associated with a number of biological alterations within the tumor itself and its microenvironment. The degree of tissue hypoxia and molecular phenotypes are major factors influencing tissue radio-sensitivity [5,8,9,10,11,12]. Hypoxia in tumor tissue can be reflected by direct measurement of tissue oxygen pressure through the microelectrode method or by detection of hypoxia-inducible factor 1α (HIF-1α) expression [13], or by imaging methods in vitro, such as by positron emission tomography (PET) with the hypoxia tracers [14]. However, these prediction methods are often limited by the difficulty of obtaining tumor tissue, applying imaging technology, and sample processing. Different molecular phenotypes play different roles in tissue radio-sensitivity, and the overexpression of peptidyl-arginine deiminase 4 (PAD4) promoted radio-resistance, survival, migration, and invasion of NPC cells [15]; the overexpression of the long non-coding RNAs, such as plasmacytoma variant translocation 1 (PVT1) and miR-483-5p, reduce NPC radio-sensitivity by decreasing radiation-induced apoptosis and DNA damage [16]. The antidifferentiation noncoding RNA (ANCR), microRNA-124, microRNA-372, microRNA-195-3p, and miR-450b-5p were associated with high tissue radio-sensitivity [17,18,19,20,21,22]. Hence, it is hard for us to obtain a comprehensive assessment from just one or two molecular phenotypes. Additionally, the detection of tumor molecular phenotypes is usually too expensive and time-consuming to be widely used in clinical practice. Therefore, it would be highly desirable for clinicians to find easily accessible and practical indicators to predict radio-sensitivity.

The peripheral blood transferrin, i.e., TRF, is one of the nutrition indicators that are widely tested in clinical practice, which can reflect the synthesis capacity of the liver and the metabolism of iron ions. Zhang Z et al. found that radiation-induced cell death was iron-dependent in triple-negative breast cancer. Cells being exposed to an iron-saturated microenvironment will cause more death while silencing the expression of transferrin reduced radiation-induced cell death [23]. A study of nutritional intervention for NPC patients found that patients who were insensitive to IMRT had lower levels of transferrin before IMRT. However, it did not further study the relationship between transferrin and radio-sensitivity or prognostic outcomes [24]. In addition, according to some studies, patients with a lower level of baseline TRF had worse outcomes for rectal, liver, and esophageal cancers [25,26,27]. In the existing basic and clinical studies, the TRF has shown its potential predictive value for prognosis and radio-sensitivity in some types of cancer. Therefore, we speculate that the level of TRF before IMRT in NPC patients may be related to radio-sensitivity and survival. To the best of our knowledge, there are few reports to this effect. This study aims to evaluate whether TRF levels before IMRT have predictive value for radio-sensitivity and outcome in NPC patients.

## 2. Materials and Methods

### 2.1. Ethical Approval of the Study Protocol

The study protocol was approved by the Ethics Review Board for Human Research of The First Affiliated Hospital of Fujian Medical University (MTCA, ECFAH of FMU [2015]084-1).

### 2.2. Criteria

Inclusion criteria: (1) pathologically diagnosed as NPC; (2) receiving radical IMRT; (3) aged 18–80 years old (inclusive). Exclusion criteria: (1) with history of other malignant tumors; (2) lack of serum transferrin data before radiotherapy; (3) failing to complete IMRT as planned; (4) obtaining no nasopharyngeal magnetic resonance image (MRI) or/and electronic nasopharyngoscope one month after IMRT; (5) with diseases affecting transferrin levels (e.g., acute hepatitis, cirrhosis, nephrotic syndrome, iron deficiency, or anemia).

### 2.3. Data Collection

All data were groomed with a unified data collection form. Basic information included age, gender, chronic disease (diabetes, hypertension, heart disease, cerebrovascular disease, and hyperlipemia), and education background. Clinical data included WHO histology type, TNM stage (restaged according to the eighth edition of Union for International Cancer Control/American Joint Committee on Cancer) [6], main treatments (absorbed dose of GTVnx, chemotherapy or not and the regimen of chemotherapy), laboratory examination results, image examination reports, and radio-sensitivity data (pathology report, MRI reports).

### 2.4. Detection of TRF

Instant ELISA kits were used to measure the concentration of serum TRF. The main principle of this detection is to bind the TRF with the anti-human TRF antibody. After the formation of antigen-antibody complex, the sandwich is formed by the addition of the second (detector) antibody, and a substrate solution is added that reacts with the enzyme-antibody-target complex to produce measurable signal. The calculation of the concentration of TRF in serum was acquired by testing the OD value of the complex with the apparatus ultraviolet spectrophotometer and comparing the results with the standard curve.

### 2.5. Treatments

All patients received IMRT, which was administered with accelerators. The primary nasopharynx (gross tumor volume of nasopharynx, GTVnx) and cervical lymph node metastases (gross tumor volume of node, GTVnd) were determined based on the results of imaging and endoscopy. The “high-risk clinical tumor volume” (CTV1) was defined as the area from 0.5 to 1.0 cm around the GTVnx and included the areas of nasopharyngeal mucosa and 0.5 cm below the nasopharyngeal mucosa. The “low-risk clinical target volume” (CTV2) was defined as the margin from 0.5 to 1.0 cm around CTV1 and the draining area of lymph nodes. The planning target volumes—(PTV)-nx, PTVnd, PTV-1, and PTV-2—were constructed by expanding GTVnx, GTVnd, CTV1, and CTV2, respectively, by 3 mm. The prescribed dose (in Gy) for GTVnx, GTVnd, CTV1, and CTV2 was 66 to 76, 66 to 70, 60 to 62, and 50 to 56/30 to 33Fx, respectively, for six to seven weeks, and it was administered at five fractions per week.

Organ at risk dose limitation was performed according to our department’s treatment routine [28]. Patients treated with neoadjuvant or adjuvant chemotherapy received 5-fluorouracil (500–750 mg/m^2^/d) from day 1 through day 5 for five days, as well as cisplatin (75–80 mg/m^2^) on days 1 to 3, or paclitaxel (135–175 mg/m^2^) on day 1, or cisplatin (75–80 mg/m^2^) on days 1 to 3, or docetaxel (75 mg/m^2^) on day 1 and cisplatin (75–80 mg/m^2^) on days 1 to 3, every 3 weeks. With regard to concurrent chemotherapy, patients received cisplatin (40 mg/m^2^) weekly or 80 mg/m^2^ every three weeks.

### 2.6. Assessment of Radio-Sensitivity

According to the treatment routine of our department, one month after radiotherapy, electronic nasopharyngoscope and MRI were used for a comprehensive evaluation [28]. (1) Radiation sensitivity: the local tumor lesions disappeared completely in the nasopharyngeal MRI scan, and no tumor residues were found under the electronic nasopharyngoscope. (2) Radiation insensitivity: the nasopharyngeal MRI scan revealed residual nasopharyngeal tumor lesions, or nasopharyngeal endoscopy found residual tumors, or rough nasopharyngeal mucosa was confirmed by pathological examination of biopsy as a residual tumor.

### 2.7. Follow-Up

All patients received follow-ups every three months during the first three years, every six months during years 4 and 5, and annually in the coming years. Each follow-up comprised a complete physical examination, an electronic nasopharyngoscopy or an indirect nasopharyngoscopy, biochemical profiles, MRI of the head and neck, CT of the chest, ultrasound of the abdomen, and Epstein-Barr virus (EBV) serology. For each follow-up, further examinations were arranged if clinically necessary. The final follow-up was on 1 October 2019.

### 2.8. Statistical Analyses

The SPSS 26 software was used for analysis. The chi-square test was used to compare the count data between the groups. Univariate and multivariable logistic regression analysis models were used to analyze the correlation between TRF before IMRT and radio-sensitivity, and the Kaplan-Meier method and multivariate Cox regression analysis model were utilized to analyze the influence of TRF before IMRT on the overall survival. A two-sided test was used, and *p* < 0.05 was considered statistically significant.

## 3. Results

From October 2012 to October 2016, a total of 355 patients were included in the study, of which seven patients were excluded due to their missing MRI examination. A total of 348 patients were assessed in the analysis, including 248 males and 100 females. They ranged in age from 18 to 78 years old, with a median age of 50 years old. An amount of 83 patients (23.8%) were 40 years old and younger; 265 (76.2%) patients were over 40. An amount of 70 patients (20.1%) received more than 70 Gy irradiation, and the other patients (79.9%) received irradiation of 70 Gy and below. The absorbed dose of GTVnx was 69.55 ± 2.05 Gy. According to AJCC 8th staging system, the proportions of patients with T stages 1, 2, 3, and 4 were 9.5%, 22.4%, 34.5%, and 33.6%, respectively. The proportion of patients with N0, N1, N2, and N3 were 38.8%, 11.8%, 35.1%, and 14.4%, respectively. Seven patients were in clinical stage I, 28 patients were in stage II, 148 patients were in stage III, and 165 patients were in stage IV. The level of TRF before IMRT ranged from 1.34 g/L to 3.89 g/L, median was at 2.16 (1.90~2.42) g/L, and mean was at (2.20 ± 0.42) g/L. Patients were divided into high-level and low-level groups based on the median value of TRF level (2.16 g/L). There were 175 patients in the low-level TRF group (LTRF), and 173 patients were in the high-level TRF group (HTRF). The chi-square test was used to analyze the differences of included factors between the LTRF and HTRF groups. There was also no significant difference in T stage, N stage, clinical TNM stage, histology type, age, radiation dose, chronic disease, degree of education, and chemotherapy between the two groups. However, there was a significant difference in genders (*χ*^2^ = 2.283, *p <* 0.0001) between the LTRF and HTRF groups. Basic information is shown in Table 1.

Of the 348 NPC patients, 242 (69.5%) were radio-sensitive, and 106 (30.5%) were radio-insensitive. The radio-sensitivity rates of the LTRF group and HTRF were 64.6% and 74.6%, respectively (*χ*^2^ = 4.103, *p* = 0.043). Univariate logistic regression was carried out for these 348 patients with NPC to investigate the effects of TRF before IMRT (>2.16 g/L vs. ≤2.16 g/L), age (≤40 years vs. >40 years), gender (male vs. female), histology type (I–II vs. III), with chronic disease (no vs. yes), T classification (T1-2 vs. T3-4), education level (literate vs. illiterate), chemotherapy (no vs. yes), and absorbed dose of GTVnx (>70 Gy vs. ≤70 Gy) on radio-sensitivity. Univariate logistic regression analysis showed that the level of TRF before IMRT was associated with radio-sensitivity (*OR* = 1.609, *p* = 0.044). Unfortunately, gender (*OR* = 1.107, *p* = 0.692), age (*OR* = 1.102, *p* = 0.939), degree of education (*OR* = 1.430, *p* = 0.149), with chronic disease or not (*OR* = 0.621, *p* = 0.103), chemotherapy or not (*OR* = 1.246, *p* = 0.508), T stage (*OR* = 1.121, *p* = 0.651), and histology type *(OR* = 0.352, *p* = 0.766) showed no significant correlation with radio-sensitivity. Multiple logistic regression analysis found that the level of TRF (*OR* = 1.702, *p* = 0.033) was an independent factor influence radio-sensitivity (Table 2). In the multivariate logistic regression analysis, gender, age, degree of education, with chronic disease or not, chemotherapy or not, T stage, histology type, and absorbed dose of GTVnx showed no correlation with radio-sensitivity.

Follow-up ended on 1 October 2019. The median follow-up was 51.7 months, and the rate of follow-up was 96.8%. The median duration of survival was 51.2 months. The 5-year OS was 78.4% for these 348 patients with NPC, 84.4% for the HTRF group, and 73.1% for the LTRF. Kaplan-Meier and log-rank tests showed that patients in the LTRF achieved a significantly worse OS (*χ*^2^ = 5.388, *p* = 0.02) than those in the HTRF (Figure 1).

Multivariate Cox proportional hazard regression was carried out for these 348 patients with NPC to investigate the effects of TRF before IMRT (>2.16 g/L vs. ≤2.16 g/L), age (≤40 years vs. >40 years), gender (male vs. female), with chronic disease (no vs. yes), education level (literate vs. illiterate), chemotherapy (no vs. yes), T classification (T1-2 vs. T3-4), histology type (I–II vs. III), and absorbed dose of GTVnx (>70 Gy vs. ≤70 Gy) on OS. In this analysis, we found that the low level of TRF before IMRT was an independent prognostic factor for worse OS of patients with NPC (*HR* = 1.706; 95% *CI* 1.065~2.731; *p* = 0.026). Besides, T stage and the degree of education level also showed significant difference on OS. The other factors included gender, age, with chronic disease or not, chemotherapy or not, histology type, and absorbed dose of GTVnx, which did not exhibit statistical significance in predicting OS of patients with NPC (Table 3).

## 4. Discussion

In this study, patients in the LTRF group had lower radio-sensitivity and worse OS. In conclusion, low level of TRF before IMRT was a risk factor for radio-sensitivity and a prognostic factor for poor OS in NPC patients. Based on our results, the baseline TRF may serve as a new clinical indicator to predict outcomes in NPC patients.

However, the specific mechanism by which TRF affects radio-sensitivity and OS remains unclear. Ferroptosis is considered as a novel mechanism that affects tumor radiation response. It has been reported that ferroptosis is involved in improving radio-sensitivity by regulating the functional expression of tumor-mutated gene p53, promoting the generation of reactive oxygen species (ROS), malondialdehyde, downregulating HIF-1α expression, and ameliorating the hypoxia microenvironment [23,29,30]. Guang Lei et al. found that ionizing radiation induced ferroptosis in cancer cells. They also found that increased ferroptosis correlated with better response and longer survival in esophageal cancer patients receiving radiotherapy [31]. These studies show that ferroptosis influences not only radio-sensitivity, but also survival in cancer patients. Ferroptosis is an iron-dependent form of regulated cell death caused by intracellular unrestricted lipid peroxidation and subsequent membrane damage. Shigeta S et al. mentioned that increasing intracellular iron levels would promote ferroptosis [30]. The transmembrane transport of iron ions is achieved through the binding of extracellular TRF to TRF receptors (TfRs) located in the cell membrane. Gao M et al. demonstrated that TRF and TfRs were required for ferroptosis. Deleting transferrin or the transferrin receptor would inhibit ferroptosis [32]. It has been reported that the high expression of TRF or TfRs will promote the occurrence of ferroptosis [29,33]. Based on the above principles, we speculate that, for NPC patients who received radiotherapy, the high level of TRF will cause more incidence of ferroptosis that influences radio-sensitivity and OS.

Besides our study, other studies also found that the levels of TRF or TfRs were prognostic factors for patients with malignant tumors [27,34,35,36]. It was reported that, for colorectal cancer patients who were in Stage I-III, the pre-operative level of TRF was independently associated with shorter cancer-specific survival (CSS) [26]. Another study of baseline TRF in patients with esophageal cancer found that low preoperative transferrin levels were strongly correlated with poor OS [27]. In addition to TRF, the level of TfRs was also found to be related to the outcomes of cancers [35,36]. Wu, H. and his colleagues found that the low level of TfR1 was an independent risk factor for OS in patients with osteosarcoma. Zhao, Q. F. et al. found that the low expression of TfR2 predicted shorter OS in gastric cancer patients, and it was significantly correlated with TNM stage of patients [36]. In our study, the results showed that a low level of TRF before IMRT in patients with NPC is associated with worse OS and radio-sensitivity. Therefore, we believe that low TRF level before radiotherapy is a negative factor for clinical prognosis. According to the current literature survey, the correlation between transferrin and radio-sensitivity is rarely reported in NPC patients. Therefore, this is the main innovation of this paper. However, as a retrospective study, we were unable to obtain the expression level of TfRs in tumor tissues to further assess whether it was closely related to prognosis. In addition, TRF is also an acute reactant that is closely related to inflammatory states.

Many clinical studies have found that, for patients with head and neck cancers, those who have a good nutritional state before radiotherapy or accept early nutritional intervention can receive a better quality of life and longer survival time [37,38,39,40,41]. Our previous research found that patients with good nutritional status before radiotherapy had higher five-year OS than those with poor nutritional status (61.8% vs. 77.1%, *p* = 0.02) [39]. TRF, as one of the commonly used indicators for clinical evaluation of nutritional status, has the characteristics of being convenient, simple, and fast-testing. Several articles have reported that the baseline level of TRF in tumor patients is significantly correlated with tumor stage, prognosis, or quality of life [26,27,42]. Yamane, T. et al. found that, in patients with esophageal cancer, low preoperative TRF levels were closely associated with poorer performance status, late T staging, and short postoperative OS [27]. Sawayama, H. et al. found that, in patients with rectal cancer, low preoperative TRF level was closely related to short postoperative OS and cancer-specific survival (CSS), and preoperative low TRF was associated with low hemoglobin, low albumin, and high white blood cell counts, as well as high C-reactive protein of pre-operation [26].

One thing that needs to be pointed out is that the five-year OS of our whole group was 78.4%, which was slightly lower than similar clinical reports. According to the latest released paper of the Cancer Center at Sun Yat-Sen University, the five-year OS of NPC patients in a multicenter study of 16 hospitals was 83.4% [43], and the ratio of similar clinical studies was also more than 80% [4,43,44]. The main reason is that about 90% of our patients were in clinical stage III or stage IV, with 49.7% in stage IV, which was much higher than that in those similar studies. The other thing we need to illustrate is that the result of radio-sensitivity to predict OS was not statistically significant in the univariate survival analysis. According to our further analysis, this may be due to the fact that patients with poor response or resistance to radiotherapy often receive adjuvant chemotherapy or maintenance chemotherapy, which may improve survival. Another possible reason is that patients with high radio-sensitivity may have a higher rate of distant metastasis, which will increase the mortality of patients. However, limited by this study is a retrospective study, as well as a lack of relevant data to explain this phenomenon. Further research is needed to confirm this.

Our study has some limitations. The primary one was that it was a single-center retrospective study, and not all laboratory and imaging results were available for all patients. Another one was that, although there were some possible mechanisms for the effect of transferrin on radio-sensitivity and prognosis, they had not been confirmed by the results of experimental studies. To further verify our conclusion, we can initiate some experimental studies to explore the complex connection among TRF, TRF receptors, and ferroptosis. In addition, multicenter observational and interventional clinical studies are needed to further verify the relationship between TRF and radio-sensitivity, as well as survival in patients with NPC.

## 5. Conclusions

In conclusion, NPC patients with higher levels of TRF before IMRT are more likely to be radio-sensitive and have better OS. It is suggested that the level of TRF may be used as a predictor of radio-sensitivity and OS in patients with NPC in clinical practice.

## Figures and Tables

**Figure 1 jpm-13-00511-f001:**
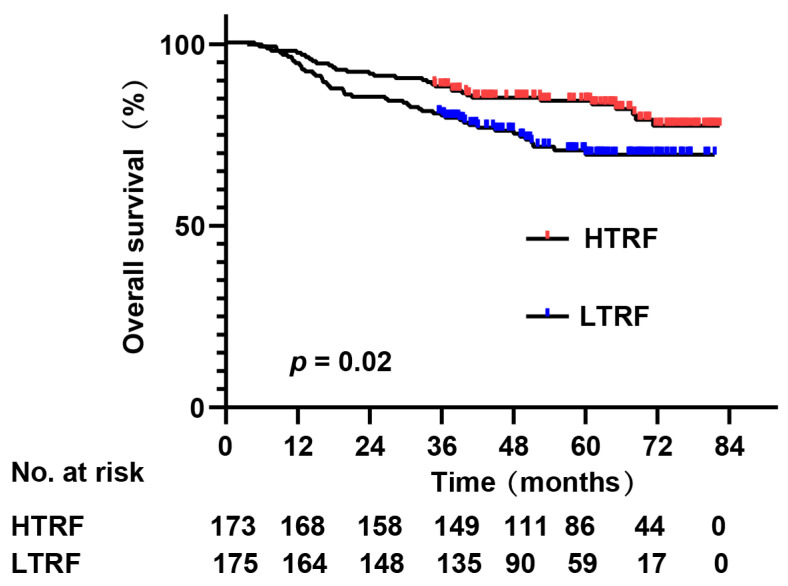
Kaplan-Meier estimate of overall survival (OS) according to the level of TRF before IMRT.

**Table 1 jpm-13-00511-t001:** Baseline data of NPC patients with different levels of TRF before IMRT.

Variables	LTRF (≤2.16 g/L)	HTRF (>2.16 g/L)	*χ* ^2^	*p*
Gender				
Male	143	105	18.772	0.001
Female	32	68		
Age(years)				
≤40	40	43	0.191	0.662
>40	135	130		
Education				
Illiteracy	67	64	0.062	0.804
Not illiteracy	108	109		
Chronic disease				
No	139	127	1.749	0.186
Yes	46	36		
Histology type *				
I + II	38	42	0.323	0.570
III	137	131		
AJCC 8th T classification				
T1–T2	51	60	1.229	0.268
T3–T4	124	113		
AJCC 8th N classification				
N0–N1	97	79	3.318	0.069
N2–N3	78	94		
AJCC 8th TNM classification				
I–III	85	90	2.276	0.131
IV	98	75		
Chemotherapy				
No	19	27	1.711	0.191
Yes	156	146		
Absorbed dose of GTVnx (Gy)				
≤70	142	135	0.518	0.472
>70	32	38		

Abbreviations: GTVnx, gross tumor volume of nasopharynx; IMRT, intensity-modulated radiation therapy; TRF, serum transferrin; NPC, nasopharyngeal carcinoma. Histology type *: type I = keratinizing squamous cell carcinoma; type II = differentiated nonkeratinizing carcinoma; type III = undifferentiated nonkeratinizing carcinoma.

**Table 2 jpm-13-00511-t002:** Multivariate logistic analysis of the risk factors of radio-sensitivity in NPC patients.

Variables	Assign	B	SE	Wald	*p*	OR (95%CI)
Gender	Male					
	Female	0.305	0.281	1.059	0.278	1.356 (0.782–2.352)
Age (years)	≤40					
	>40	0.190	0.294	0.419	0.517	1.209 (0.680–2.150)
Education	Illiteracy					
	Not illiteracy	0.450	0.266	2.860	0.091	1.568 (0.931–2.641)
Chronic disease	No					
	Yes	0.446	0.311	2.056	0.152	1.561 (0.849–2.871)
Histology type *	I + II					
	III	0.233	0.291	0.639	0.424	1.262 (0.713–2.234)
AJCC 8th T classification	T1–T2					
	T3–T4	0.110	0.261	0.176	0.675	1.116 (0.669–1.862)
Chemotherapy	No					
	Yes	0.168	0.362	0.216	0.642	1.183 (0.582–2.406)
Absorbed dose of GTVnx (Gy)	<70					
	≥70	0.197	0.293	0.453	0.501	1.218 (0.686–2.163)
Transferrin before IMRT	LTRF					
	HTRF	0.532	0.249	4.549	0.033	1.702 (1.044–2.775)

Abbreviations: GTVnx, gross tumor volume of nasopharynx; IMRT, intensity-modulated radiation therapy; TRF, serum transferrin; NPC, nasopharyngeal carcinoma. Histology type *: type I = keratinizing squamous cell carcinoma; type II = differentiated nonkeratinizing carcinoma; type III = undifferentiated nonkeratinizing carcinoma.

**Table 3 jpm-13-00511-t003:** COX regression analysis results of OS for patients with nasopharyngeal carcinoma.

Variables	Assign	B	SE	Wald	*p*	HR (95%CI)
Gender	Male					
	Female	0.119	0.306	0.151	0.697	1.126 (0.619–2.051)
Age(years)	≤40					
	>40	0.213	0.228	0.548	0.459	1.238 (0.704–2.176)
Education level	Illiteracy					
	Not illiteracy	0.563	0.264	4.535	0.033	1.756 (1.046–2.949)
With chronic disease	No					
	Yes	0.476	0.258	3.421	0.064	1.610 (0.972–2.667)
Histology type *	I + II					
	III	0.121	0.266	0.206	0.650	1.128 (0.670–1.899)
AJCC 8th T classification	T1–T2					
	T3–T4	1.024	0.318	10.369	0.001	2.784 (1.493–5.192)
Chemotherapy	No					
	Yes	0.210	0.328	0.410	0.522	1.234 (0.649–2.346)
Absorbed dose of GTVnx (Gy)	<70					
	≥70	0.285	0.260	1.195	0.274	1.329 (0.798–2.214)
Transferrin before IMRT	LTRF					
	HTRF	0.534	0.240	4.945	0.026	1.706 (1.065–2.731)

Abbreviations: GTVnx, gross tumor volume of nasopharynx; IMRT, intensity-modulated radiation therapy; TRF, serum transferrin; NPC nasopharyngeal carcinoma. Histology type *: type I = keratinizing squamous cell carcinoma; type II = differentiated nonkeratinizing carcinoma; type III = undifferentiated nonkeratinizing carcinoma.

## Data Availability

The data that support the findings of this study are available upon request from the corresponding author. The data are not publicly available due to privacy or ethical restrictions.

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
