# Peer review of "Prognostic Value of Serum Transferrin Level before Radiotherapy on Radio-Sensitivity and Survival in Patients with Nasopharyngeal Carcinoma"

_jpm, 2023, doi:10.3390/jpm13030511_

Round 1
Reviewer 1 Report
Comments and Suggestions for Authors
Given the reality that nonresponse to radiotherapy remains an obstacle to successful treatment in nasopharyngeal carcinoma (NPC), there are several studies reporting that blood transferrin (TRF) level has showed its potential value in prognosis for radiosensitivity in some types of cancers, including NPC. Authors in this manuscript further investigated the correlation in NPC patients. Important, they revealed that TRF could be potentially a prognostic factor for overall survival in NPC patients. Based on the quality of data presentation and scientific soundness, my overall recommendation is to accept after minor revision. Please address the following issues before acceptance.
Introduction
· NPC is unique in the aspects of epidemiology, pathology, and clinical manifestation. High infection with Epstein-Barr virus (EBV) in NPC is one of the important characteristics. Please provide sufficient background and include all relevant references, when it comes to the topic about the roles of molecular phenotypes on radiosensitivity (Page 5, Line 4).
· As authors mentioned, TNM stage is recognized as the best predictor of radiosensitivity (Page 4, Line 18). Please provide sufficient background and include all relevant references.
· Please also confirm the goal in this study was to investigate the potential of TRF as an independent biomarker or a supplement of TNM staging (Page 5, Line 12). If it is the latter case, please show the evidence to demonstrate the additional benefit by TRF in addition to TNM stages.
Materials and Methods
· Please add the method of TRF measurement in clinic. It would be essential information for comparing the results from different studies.
Results
· Overall, the results are not adequately described, which must be improved. There is lack of description about the association of TRF with clinicopathological parameters in Table 1, the risk factors except for TRF to radiosensitivity in Table 2, as well as that to OS in Table 3. For example, although the statistics is clearly presented in Table 2, the observation of the correlation between histologic grade/ T classification and radiotherapy needs to be fully described and discussed in the text, to provide with a full picture of the messages concluded from the dataset.
· Please add the legend for “Histologic grade*” across all tables. What does it mean by “*”?
Others
1. (Page 3, Line 9) Please add the information about "our center".
2. (Page 3, Line 18) Authors performed Cox proportional hazard regression to investigate the correlation between TRF level and radiosensitivity. Data should be presented in a certain way like (HR: X; 95% CI: X, X; P < X), instead of showing the P value.
3. (Page 4, Line 12) Please clarify the statement of “The difference in radiosensitivity is one of the main reasons”. I assume that radiosensitivity is one of the clinical outcomes.
4. (Page 12, Line 3) Please clarify the statement of “According to the current literature survey, we have not found any articles on the direct study of transferrin and radiosensitivity. Therefore, this is the main innovation of this paper”. As author cited the publication of J Healthc Eng. 2022;2022: 2531671., the correlation of TRF to radiosensitivity has been previously reported.
5. Please confirm reference 24.
Author Response
Dear Reviewer,
Thank you for your review and valuable and helpful comments on our manuscript. We have carefully revised the manuscript according to your comments. Revised sections are made under tracked change and marked in red. Point-by-point responses to the editors and reviewers’ comments are listed below this letter.
We do hope that the revised version of the manuscript is now acceptable for publication in your journal. If you have any queries, please don’t hesitate to contact us.
Thank you again for your valuable comments and suggestions. I look forward to hearing from you soon.
With best wishes.
Yours sincerely,
Professor Jin-Sheng Hong
Point-by-point response to the comments of the Reviewer:
Introduction
1. NPC is unique in the aspects of epidemiology, pathology, and clinical manifestation. High infection with Epstein-Barr virus (EBV) in NPC is one of the important characteristics. Please provide sufficient background and include all relevant references, when it comes to the topic about the roles of molecular phenotypes on radiosensitivity (Page 5, Line 4).
Response: Thank you for your careful review of our paper. About the roles of molecular phenotypes on radiosensitivity, there were numbers of paper reported different molecular phenotypes would influence radiosensitivity of NPC. We have made corresponding supplements in Introduction part (Page 5, Line 12-20) and Reference part (Page 16, Lines 13-34)
2. As authors mentioned, TNM stage is recognized as the best predictor of radiosensitivity (Page 4, Line 18). Please provide sufficient background and include all relevant references.
Response: Thank you for your careful review of our paper. We apologize for the typographical error and inaccurate expression in " TNM stage is recognized as the best predictor of radiosensitivity " here. We deleted this sentence in the text.
3. Please also confirm the goal in this study was to investigate the potential of TRF as an independent biomarker or a supplement of TNM staging (Page 5, Line 12). If it is the latter case, please show the evidence to demonstrate the additional benefit by TRF in addition to TNM stages.
Response: T stage depends on imaging examination, and TRF as a hematological examination is another form of predicting radiosensitivity. The objective of this study was to investigate the potential of TRF as an independent predictor of radiosensitivity and an independent prognostic factor of OS, so we included TRF level and T stage in multiple logistics regression analysis and multiple Cox regression analysis. For the expression of “……as a supplement of TNM staging.”, we apologize for the inaccurate expression and we have made corresponding revisions in the revised manuscript. (Page 6, Line 2-3).
Materials and Methods
Please add the method of TRF measurement in clinic. It would be essential information for comparing the results from different studies.
Response: Thank you for your suggestion. We have added the method of TRF measurement in our hospital in Materials and Methods part (Page 7, Line 20).
Results
1. Overall, the results are not adequately described, which must be improved. There is lack of description about the association of TRF with clinicopathological parameters in Table 1 the risk factors except for TRF to radiosensitivity in Table 2, as well as that to OS in Table 3. For example, although the statistics is clearly presented in Table 2, the observation of the correlation between histologic grade/ T classification and radiotherapy needs to be fully described and discussed in the text, to provide with a full picture of the messages concluded from the dataset.
Response: Thank you for your suggestion. We have improved the description of our results in Results part ((Page 10, Line 1-7); (Page 10, Line 11-22); (Page 11, Line 1-4); (Page 10, Line 18-20)).
2. Please add the legend for “Histologic grade*” across all tables. What does it mean by “*”?
Response: The symbol of “*” act as a reminder to indicate footnotes to the table. We have added legend for "Histologic grade*" in the footnote (revised table1-3)
Others
1. (Page 3, Line 9) Please add the information about "our center".
Response: We have added the information about "our center " in revised manuscript (Page 3, Line 13; Page 15, Line 6-7).
2. (Page 3, Line 18) Authors performed Cox proportional hazard regression to investigate the correlation between TRF level and radiosensitivity. Data should be presented in a certain way like (HR: X; 95% CI: X, X; P < X), instead of showing the P value.
Response: Thanks for your suggestion. We have made revisions based on your suggestion. (Page 4, Line 3-6)
3. (Page 4, Line 12) Please clarify the statement of “The difference in radiosensitivity is one of the main reasons”. I assume that radiosensitivity is one of the clinical outcomes.
Response: Thank you for your careful review of our paper. Radiosensitivity can indeed reflect the short-term efficacy of patients receiving radiotherapy. However, from the perspective of radiobiology, radiosensitivity refers to the degree of response of tumor tissue or cells to irradiation. Differences in radiosensitivity may lead to differences in the efficacy of patients.
4. (Page 13, Line 3) Please clarify the statement of “According to the current literature survey, we have not found any articles on the direct study of transferrin and radiosensitivity. Therefore, this is the main innovation of this paper”. As author cited the publication of J Healthc Eng. 2022;2022: 2531671., the correlation of TRF to radiosensitivity has been previously reported.
Response: Thank you for your careful review. This study that we cited only showed a correlation between TFR and radiosensitivity using a t-test, but the specific relationship is not further elucidated. Our statement is inappropriate and has been revised (Page 13,Line 20).
5. Please confirm reference 24.
Response: This paper is a doctoral thesis. We have removed this reference from the revised manuscript.

Reviewer 2 Report
This is a well-designed study investigating the impact of serum transferrin before IMRT on radiosensitivity and overall survival in patients with nasopharyngeal carcinoma.
However, there are still some issues including the "cutoff value of TRF" that need further major revision of this manuscript.
1. In the Statistical analyses of Materials and Methods, the authors stated that “the ROC curve was used to find the cutoff point”. However, the results of ROC analysis were not presented in this manuscript.
It is important to perform the ROC analysis for the cutoff values of the serum transferrin (TRF) level in this study.
2. Moreover, the grouping of high-level and low-level of TRF based on the “cutoff value of TRF determined by ROC analysis” can be a better indicator than the median value of TRF in this study.
3. As a results, the grouping of TRF based on “the cutoff value of TRF determined by ROC analysis” should be utilized in the subsequent analyses including the Kaplan-Meier estimate of overall survival, univariate, multivariate logistic regression, and COX regression analysis (Table 2-3, and Fig 1).
4. Radiosensitivity plays a vital role in treatment response and outcomes of NPC patients. Although univariate and multivariate logistic regression were performed for the radiosensitivity in NPC patients, the impact of radiosensitivity on OS was not performed in the COX regression analysis.
Therefore, further analysis of radiosensitivity in the Kaplan-Meier estimate of overall survival and COX regression analysis (Table 3) is recommended.
5. In the Discussion, please further discuss the differences between the grouping of TRF based on the “median value of 2.16g/L” and the “cutoff value of TRF determined by ROC analysis”.
Author Response
Dear Reviewer,
Thank you for your review and valuable and helpful comments on our manuscript. We have carefully revised the manuscript according to your comments. Revised sections are made under tracked change and marked in red. Point-by-point responses to the editors and reviewers’ comments are listed below this letter.
We do hope that the revised version of the manuscript is now acceptable for publication in your journal. If you have any queries, please don’t hesitate to contact us.
Thank you again for your valuable comments and suggestions. I look forward to hearing from you soon.
With best wishes.
Yours sincerely,
Professor Jin-Sheng Hong
Point-by-point response to the comments of the Reviewer:
1. In the Statistical analyses of Materials and Methods, the authors stated that “the ROC curve was used to find the cutoff point”. However, the results of ROC analysis were not presented in this manuscript. It is important to perform the ROC analysis for the cutoff values of the serum transferrin (TRF) level in this study. Moreover, the grouping of high-level and low-level of TRF based on the “cutoff value of TRF determined by ROC analysis” can be a better indicator than the median value of TRF in this study.
2. As a results, the grouping of TRF based on “the cutoff value of TRF determined by ROC analysis” should be utilized in the subsequent analyses including the Kaplan-Meier estimate of overall survival, univariate, multivariate logistic regression, and COX regression analysis (Table 2-3, and Fig 1).
Response: Thank you for your careful review of our paper. For the above questions, we make a general explanation as follows. In this study, we used R language survival ROC analysis package to calculate the cut-off value of TRF on survival, and found that the best cut-off value was 1.34g/L, which was the minimum value in TRF data. however, the area under ROC curve was 0.566. Therefore, ROC analysis is not suitable for this study to take the cut-off value. In the “8. statistical analysis” section, we have removed the description of the ROC curve.
3. Radiosensitivity plays a vital role in treatment response and outcomes of NPC patients. Although univariate and multivariate logistic regression were performed for the radiosensitivity in NPC patients, the impact of radiosensitivity on OS was not performed in the COX regression analysis. Therefore, further analysis of radiosensitivity in the Kaplan-Meier estimate of overall survival and COX regression analysis (Table 3) is recommended.
Response: Thank you for your advice. We attempted to include radiosensitivity in the Cox proportional hazard regression model to analyze the influence of radiosensitivity on OS, but the results did not show that radiosensitivity was an independent prognostic indicator of OS. This may be due to the fact that patients who respond poorly to radiotherapy or are resistant to radiotherapy often receive adjuvant chemotherapy or maintenance chemotherapy, which may affect patient survival.
4. In the Discussion, please further discuss the differences between the grouping of TRF based on the “median value of 2.16g/L” and the “cutoff value of TRF determined by ROC analysis”.
Response: Thank you for your careful review of our paper. We are very sorry for our carelessness in writing, we have explained the reasons why ROC is not applicable in the response of question1-2.

Round 2
Reviewer 2 Report
The revised manuscript has been much improved.
The responses form the authors are also adequate.
Author Response
Dear Reviewer,
Thank you for your review and valuable and helpful comments on our manuscript. We have carefully revised the manuscript according to your comments. Revised sections are made under tracked change and marked in red.
We do hope that the revised version of the manuscript is now acceptable for publication in your journal. If you have any queries, please don’t hesitate to contact us.
Thank you again for your valuable comments and suggestions. I look forward to hearing from you soon.
With best wishes.
Yours sincerely,
Professor Jin-Sheng Hong
Response: Thank you for your careful review of our paper. In our study, we found that low level of TRF before IMRT was a risk factor for radio-sensitivity and a prognostic factor for poor OS in NPC patients. However, we found that there was no statistical influence of radio-sensitivity on OS. According to our further analysis, this may be due to the fact that patients with poor response or resistance to radiotherapy often receive adjuvant chemotherapy or maintenance chemotherapy, which may improve survival. Another possible reason is that patients with high radio-sensitivity may have a higher rate of distant metastasis which will increase the mortality of patients. However, limited by this study is a retrospective study, we are lack of relevant data to explain this phenomenon. Further research is needed to confirm this.
